# GAF-CaMP3–sfGFP, An Enhanced Version of the Near-Infrared Genetically Encoded Positive Phytochrome-Based Calcium Indicator for the Visualization of Neuronal Activity

**DOI:** 10.3390/ijms21186883

**Published:** 2020-09-19

**Authors:** Oksana M. Subach, Fedor V. Subach

**Affiliations:** National Research Center “Kurchatov Institute”, 123182 Moscow, Russia

**Keywords:** genetically encoded calcium indicator, protein engineering, calcium imaging, bacterial phytochrome, GAF-CaMP3, GECI, near-infrared, fluorescent protein

## Abstract

The first generation of near-infrared, genetically encoded calcium indicators (NIR-GECIs) was developed from bacterial phytochrome-based fluorescent proteins that utilize biliverdin (BV) as the chromophore moiety. However, NIR-GECIs have some main drawbacks such as either an inverted response to calcium ions (in the case of NIR-GECO1) or a limited dynamic range and a lack of data about their application in neurons (in the case of GAF-CaMP2–superfolder green fluorescent protein (sfGFP)). Here, we developed an enhanced version of the GAF-CaMP2–sfGFP indicator, named GAF-CaMP3–sfGFP. The GAF-CaMP3–sfGFP demonstrated spectral characteristics, molecular brightness, and a calcium affinity similar to the respective characteristics for its progenitor, but a 2.9-fold larger ΔF/F response to calcium ions. As compared to GAF-CaMP2–sfGFP, in cultured HeLa cells, GAF-CaMP3–sfGFP had similar brightness but a 1.9-fold larger ΔF/F response to the elevation of calcium ions levels. Finally, we successfully utilized the GAF-CaMP3–sfGFP for the monitoring of the spontaneous and stimulated activity of neuronal cultures and compared its performance with the R-GECO1 indicator using two-color confocal imaging. In the cultured neurons, GAF-CaMP3–sfGFP showed a linear ΔF/F response in the range of 0–20 APs and in this range demonstrated a 1.4-fold larger ΔF/F response but a 1.3- and 2.4-fold slower rise and decay kinetics, respectively, as compared to the same parameters for the R-GECO1 indicator.

## 1. Introduction

Genetically encoded calcium indicators (GECIs) are an indispensable tool for the visualization of calcium dynamics in living cells [1,2]. Recently, efforts have been made to expand the palette of available fluorescence colors of GECIs to the near-infrared region [3,4]. Near-infrared GECIs are attractive in terms of reduced light scattering and transparency of biological tissue in the infrared spectral region and provide additional colors for multiplexing florescence imaging.

Two versions of near-infrared GECIs, called NIR-GECO1 [3] and GAF-CaMP2 [4], were recently published. Both indicators are based on bacterial phytochromes in which fluorescence arises from a biliverdin (BV) chromophore. Usually, bacterial phytochromes consist of two domains, the GAF and the PAS [5]. Most of the fluorescent proteins based on BV have a two-domain structure, but several single-domain fluorescent proteins containing only the GAF domain have been developed from single-domain cyanobacteriochromes [6,7], or from two-domain *Rp*BphP1 [8]. NIR-GECO1 [3] and GAF-CaMP2 [4] were originally generated from the two-domain mIFP and the single-domain GAF-fluorescent protein (FP). Although endogenous BV is present in mammalian cells as an intermediate of heme degradation, externally added BV usually enhances the brightness of fluorescent proteins based on BV. NIR-GECO1 has excitation/emission maxima (678/704 nm), which are 36 and 30 nm far-red shifted as compared to those for GAF-CaMP2 (642/674 nm). NIR-GECO1 has been successfully applied to monitor neuronal calcium activity in cultured cells, primary neurons, and acute brain slices, and the GAF-CaMP2–superfolder green fluorescent protein (sfGFP) fusion has been used to robustly monitor calcium transient in cultured cells.

However, the first generation of bacteriophytochrome-based near-infrared GECIs have several drawbacks. Since NIR-GECO1 has an inverted response to calcium ions, it is susceptible to photobleaching under continuous illumination. Meanwhile, GAF-CaMP2–sfGFP has a positive response to calcium ions, but its ΔF/F dynamic range is limited by 78% and 169% on purified proteins and in cultured cells, respectively. However, the performance of GAF-CaMP2–sfGFP has not been verified in neurons. Both NIR-GECO1 and GAF-CaMP2–sfGFP have been unsuccessfully applied to in vivo imaging of neuronal activity in the rodent brain. Additionally, NIR-GECO1 and GAF-CaMP2–sfGFP have a low molecular brightness of 12% and 5.6% relative to GFP, respectively.

To address some of the abovementioned limitations, in this paper, we developed a version of the GAF-CaMP2–sfGFP indicator, called GAF-CaMP3–sfGFP. GAF-CaMP3–sfGFP demonstrated a 2.9- and a 1.9-fold larger ΔF/F dynamic range on purified proteins and in cultured HeLa cells, respectively. Furthermore, GAF-CaMP3–sfGFP was successfully applied to the monitoring of neuronal activity in cultured cells.

## 2. Results and Discussion

### 2.1. Improving the Dynamic Range of the Near-Infrared Bacterial Phytochrome-Based GAF-CaMP2–sfGFP Indicator in a Bacterial System

To enhance the dynamic range of the GAF-CaMP2 indicator, we performed 12 rounds of generation of random bacterial libraries, followed by their consecutive screening on Petri dishes and 96-well plates on a plate reader, as described earlier [4]. To facilitate the screening of bacterial colonies, we made a fusion of GAF-CaMP2 variants with sfGFP. During the screening procedure, we selected colonies that demonstrated the largest ΔF/F response to the elimination of calcium ion concentrations and the largest far-red fluorescence normalized to green fluorescence of the sfGFP protein. As a result of this directed selection, we found a final variant of GAF-CaMP2–sfGFP, named GAF-CaMP3–sfGFP.

GAF-CaMP3–sfGFP had 28 mutations as compared to the GAF-CaMP2–sfGFP progenitor (Figure 1 and Appendix A). Thirteen mutations were located in the fluorescent domain of GAF-CaMP3–sfGFP (Figure 2a,b); however, none of them affected the residues within the putative 4.5–6.5 Å region surrounding the BV chromophore, and therefore they could not significantly impact the fluorescent properties of BV. Mutation L111P was in the linker between the fluorescent and calcium-binding domains; we speculate that this might affect the dynamic range of the indicator. Fourteen other mutations were found in the calcium-binding domain; these mutations can presumably affect the affinity, dynamic range, and calcium ion association–dissociation kinetics of the indicator.

### 2.2. In Vitro Characterization of the Purified GAF-CaMP3–sfGFP Indicator 

First, we characterized the spectral and biochemical properties of the purified GAF-CaMP3–sfGFP indicator in vitro. Since GAF-CaMP3 alone was non-fluorescent in mammalian cells (see details below), we did not characterize it.

The spectral properties of GAF-CaMP3–sfGFP were similar to its GAF-CaMP2–sfGFP progenitor (Table 1). GAF-CaMP3-sfGFP in the apo- and sat-states has the absorption maxima characteristic for Q-band of BV chromophore in bacterial phytochromes at 640 and 654 nm, respectively (Figure 2c). In addition to NIR absorbance at the major band (called the Q-band) phytochromes also absorb at the region about 400 nm (called the Soret band) which is a characteristic band for tetrapyrroles [9]. The absorption peaks of GAF-CaMP3–sfGFP in both states at 388 and 490 nm were ascribed to the Soret band of the BV chromophore and sfGFP, respectively. GAF-CaMP3–sfGFP_sat_ had excitation/emission maxima at 648/676 nm, respectively (Figure 2d). In the apo state, these maxima were observed at 636/674 nm, respectively. The excitation/emission maxima at 498/514 nm and excitation peaks at 376 and 381 nm were attributed to sfGFP and the Soret band of the BV chromophore, respectively. GAF-CaMP3–sfGFP in the calcium-bound state had a molecular brightness (as a product of quantum yield and extinction coefficient) of 6.2% relative to the enhanced green fluorescent protein (EGFP) (Table 1). The molecular brightness of GAF-CaMP2–sfGFP vs. EGFP of 5.6% was very similar to that for GAF-CaMP3–sfGFP. Dissociation from calcium ions decreased both the quantum yield and the extinction coefficient of the GAF-CaMP3–sfGFP protein by 1.54- and 1.88-fold, respectively. In the absence and in the presence of 1 mM Mg^2+^ (the condition, mimicking cytosolic cellular Mg^2+^ concentration), GAF-CaMP3–sfGFP revealed an ΔF/F response to calcium ions of 197 and 227%, respectively (Table 1). Under the same conditions, GAF-CaMP2–sfGFP demonstrated 2.1–2.9-fold lower ΔF/F responses to calcium ions as compared to GAF-CaMP3–sfGFP (Table 1). Hence, the spectral properties and molecular brightness of the GAF-CaMP3–sfGFP purified protein were similar to those for its GAF-CaMP2–sfGFP progenitor, but purified GAF-CaMP3–sfGFP demonstrated a 2.9-fold larger ΔF/F response to calcium ions.

To estimate the possible impact of pH changes on the ΔF/F dynamic range of the GAF-CaMP3–sfGFP protein, we compared its pH stability in calcium-bound and calcium-free states. In both the sat and apo states, the far-red fluorescence of GAF-CaMP3–sfGFP changed over pH variation in the range of 3–10 (Figure 2e). Two notable transitions in the sat and apo states were found at a pH of 3.50–3.53 and 8.9–9.18, respectively (Table 1). The pH stabilities of the sat and apo states were similar but slightly different and resulted in variation in the ΔF/F dynamic range over different pH levels. Hence, variations in pH can contribute to the calcium response of the GAF-CaMP3–sfGFP indicator.

We next measured the affinity of GAF-CaMP3–sfGFP to calcium ions to confirm its applicability for the measurement of calcium transients in the cytosol of mammalian cells. Equilibrium calcium-binding titration experiments for the GAF-CaMP3–sfGFP indicator revealed a K_d_ value of 433 nM and a Hill coefficient of 1.36 (Table 1 and Figure 2f). The addition of 1 mM of magnesium ions (to simulate conditions in the cytosol of the mammalian cell) resulted in a 1.2- and 1.4-fold increase in the K_d_ and Hill coefficient values to 524 nM and 1.85, respectively (Figure 2g). Under these conditions, the calcium affinity of GAF-CaMP3–sfGFP was practically similar to that of GAF-CaMP2–sfGFP (Table 1). The K_d_ values for GAF-CaMP3–sfGFP were similar to the K_d_ values of 632 nM [4] for the GCaMP6f indicator, which robustly monitored variations of calcium ions in the cytosol of mammalian cells [12]. Hence, the GAF-CaMP3–sfGFP indicator demonstrated an affinity to calcium ions similar to that of the GAF-CaMP2–sfGFP and GCaMP6f GECIs and can be further applied for the registration of calcium ion transients in mammalian cells.

### 2.3. Calcium-Dependent Response of the GAF-CaMP3–sfGFP Calcium Indicator in HeLa Mammalian Cells

To demonstrate the applicability of the GAF-CaMP3–sfGFP indicator for the registration of calcium dynamics in mammalian cells, we optimized the conditions for its expression in HeLa cells and compared its ΔF/F response to the ionomycin-induced elevated concentration of calcium ions in HeLa cells with its GAF-CaMP2–sfGFP progenitor. We could not detect notable fluorescence in the case of GAF-CaMP3 expression alone (i.e., in the absence of sfGFP on its C-terminal end) in the HeLa cells, even in the presence of 20 µM of BV or at high levels of calcium ions upon addition of 2.5 µM ionomycin (Appendix A). At elevated concentrations of calcium ions in the HeLa cells, the brightness of the GAF-CaMP3–sfGFP fusion in the absence of externally added BV was 12-fold lower as compared to the brightness of GAF-CaMP3–sfGFP at the same conditions in the presence of 20 µM of BV (Figure 3a,b,d). The brightness of GAF-CaMP3–sfGFP in the HeLa cells was similar (*p* = 0.2287) to the brightness of the GAF-CaMP2–sfGFP progenitor. The ΔF/F response induced by ionomycin-induced calcium ion elevations in the cytosol of the HeLa cells was 1.9-fold (*p* < 0.0020) larger for GAF-CaMP3–sfGFP as compared to the same response for GAF-CaMP2–sfGFP (Figure 3c,e). Hence, the expression of GAF-CaMP3 in mammalian cells demanded its C-terminal fusion with sfGFP and the addition of 20 µM of BV; GAF-CaMP3–sfGFP demonstrated a 1.9-fold larger ΔF/F response to calcium ion elevations in mammalian cells as compared to the response for the GAF-CaMP2–sfGFP progenitor.

### 2.4. Visualization of Spontaneous and Induced Neuronal Activity in the Dissociated Culture Using the GAF-CaMP3–sfGFP Indicator and Confocal Imaging

We next attempted to demonstrate for the first time the possibility of the expression and calcium imaging of GAF-CaMP3–sfGFP in neurons and to estimate its ΔF/F responses during non-specific (spontaneous) and external electric field stimulation. To this aim, the neuronal cultures were co-transduced with recombinant adeno-associated viruses (rAAVs) carrying near-infrared NES-GAF-CaMP3–sfGFP and red NES-R-GECO1 GECIs. Similar to HeLa cells, the fluorescence of GAF-CaMP3–sfGFP was very dim in the absence of externally added BV. The addition of 10 µM of BV five or more hours before imaging resulted in clearly visible far-red fluorescence in the cytosol of cultured neurons (Figure 4a). We could monitor non-specific neuronal activity using GAF-CaMP3–sfGFP with an averaged ΔF/F response 1.3-fold larger than that of R-GECO1 (Figure 4c). The averaged rise half-times were similar for both GAF-CaMP3–sfGFP and R-GECO1 (Figure 4d); however, the averaged decay half-time for GAF-CaMP3–sfGFP was 2.3-fold longer than the decay half-time for R-GECO1 (Figure 4e). Since the decay half-time for R-GECO in the cultured neurons was 1.9-fold shorter than the decay half-time for the GCaMP6s indicator [13], the dissociation kinetics for GAF-CaMP3–sfGFP was similar to the commonly used GCaMP6s indicator. Hence, the calcium imaging of GAF-CaMP3–sfGFP in cultured neurons demanded the addition of 10 µM of BV, and GAF-CaMP3–sfGFP robustly monitored spontaneous neuronal activity with a 1.3-fold larger ΔF/F response and similar rise kinetics, but 2.3-fold slower decay kinetics as compared to the respective characteristics for the R-GECO1 indicator.

To precisely characterize the ΔF/F response and kinetics of GAF-CaMP3–sfGFP in neurons, we stimulated the neuronal cultures co-expressing the GAF-CaMP3–sfGFP and R-GECO1 indicators with an external electric field. Stimulation of the neuronal cultures co-expressing the GAF-CaMP3–sfGFP and R-GECO1 GECIs resulted in a robust response of both indicators in the cytosol of the cell (Figure 5a). The ΔF/F response of the GAF-CaMP3–sfGFP indicator in the range of 0–20 action potentials (APs) was linear (Figure 5b) and 1.4-fold larger than the response for the R-GECO1 indicator but 2.7-fold lower than GCaMP6s (Figure 5b,c). In the range of >20 Aps, the ΔF/F response for R-GECO1 was larger than that for GAF-CaMP3–sfGFP. The averaged rise and decay half-times for GAF-CaMP3–sfGFP were 1.3- and 2.4-fold slower, correspondingly, as compared to the respective time characteristics for R-GECO1 (Figure 5d,e). Hence, in the cultured neurons, GAF-CaMP3–sfGFP showed a linear ΔF/F response in the range of 0–20 APs and, in this range, demonstrated a 1.4-fold larger ΔF/F response, but 1.3- and 2.4-fold slower rise and decay kinetics, respectively, as compared to the same parameters for the R-GECO1 indicator.

## 3. Materials and Methods 

### 3.1. Mutagenesis and Library Screening

The construction of the libraries and their following screening were performed as described in reference [13] with two modifications. First, the GAF-CaMP variants were cloned into the pBAD/HisB-TorA-sfGFP plasmid using the GAF-BglII/GAF-EcoRI-r primers listed in Appendix A to express the GAF-CaMP variants in fusion with the sfGFP protein (i.e., GAF-CaMPs–sfGFP). The green fluorescence of sfGFP assisted the colony screening and potentially allowed characterization of the folding efficiency of the GAF-CaMP variants. Second, the proteins were extracted in the presence of 1 µM of BV to avoid BV wash-out.

### 3.2. Protein Purification and Characterization

Proteins were expressed and purified as described in reference [4], and 1 µM of BV was supplied during all stages of the purification and characterization.

The extinction coefficient values for the purified GAF-CaMP3–sfGFP protein in the Ca^2+^-saturated or apo states were calculated in a buffer of 30 mM of 2-[4-(2-hydroxyethyl)piperazin-1-yl]ethanesulfonic acid (HEPES), pH 7.2, 100 mM of KCl, and 1 µM of BV (buffer A) supplemented with 5 mM of CaCl_2_ or 10 mM of EDTA relative to the peak at 490 nm (for sfGFP) that had an extinction coefficient of 56,000 M^−1^ cm^−1^. Absorption spectra were registered using a NanoDrop 2000c Spectrophotometer (Thermo Scientific, Wilmington, DE, USA).

For determination of the quantum yield of purified GAF-CaMP3–sfGFP in the Ca^2+^-saturated or apo states, the integrated fluorescence values (in the range of 620–820 nm) for the protein excited at 590 nm were measured in buffer A, supplemented with 5 mM of CaCl_2_ or 10 mM of EDTA and compared with the same values for the equally absorbing at 590 nm smURFP protein (quantum yield of 17.9 [4]). Fluorescence spectra were acquired using a CM2203 spectrofluorometer (Solar, Minsk, Belarus).

pH titrations for the purified GAF-CaMP3–sfGFP protein (50 nM final concentration) in the Ca^2+^-saturated or apo states were performed in buffers of 30 mM of citric acid, 30 mM of borax, and 30 mM of NaCl, with a pH ranging from 3.0 to 10.0 supplemented with 1 µM of BV, 0.1 mM of CaCl_2_, or 0.1 mM of EDTA, respectively, incubated for 20 min at room temperature, as described in [4]. Far-red fluorescence (Ex 625 nm/Em 660–720 nm) was registered using a 96-well ModulusTM II Microplate Reader (Turner Biosystems, Sunnyvale, CA, USA).

For determination of equilibrium calcium K_d_, the fluorescence of the GAF-CaMP3–sfGFP protein (50 nM final concentration) was measured in the mixture of two stocks of buffer A containing 10 mM of EGTA or 10 mM of Ca-EGTA, as described previously [4].

### 3.3. Mammalian Plasmid Construction

In order to construct the pAAV-*CAG*-NES-GAF-CaMP3 and pAAV-*CAG*-NES-GAF-CaMP3–sfGFP plasmids, the GAF-CaMP3 and GAF-CaMP3–sfGFP genes were polymerase chain reaction (PCR)-amplified as the BglII-EcoRI and BglII-HindIII fragments, respectively, using the GAF-BglII/GAF-EcoRI-r2 and GAF-BglII/mCherry-HindIII-r primers listed in Appendix A, and then swapped with the mCherry gene in the pAAV-*CAG*-NES-mCherry vector. 

### 3.4. Mammalian Live-Cell Imaging

HeLa Kyoto cell cultures were imaged 24–48 h after transient lipofectamine transfection before and immediately after the addition of 2.5 μM of ionomycin addition using a laser spinning disk Andor XDi Technology Revolution multi-point confocal system (Andor Technology, Belfast, UK) as previously described [4]. During the transfection procedure and immediately before imaging, 20 mM HEPES, pH 7.40 and 20 µM BV were added, respectively.

### 3.5. Imaging in Primary Mouse Neuronal Cultures

The rAAV particles were purified from 10 150-cm dishes as described in the original paper [13]. Dissociated neuronal cultures were isolated from C57BL/6 mice at postnatal days 0–1 and were grown on a 24-well cell imaging black plate with a glass bottom, and then tissue culture-treated (Eppendorf, Hamburg, Germany) in Neurobasal Medium A (GIBCO, Paisley, Scotland, UK) supplemented with 2% B27 Supplement (GIBCO, Paisley, Scotland, UK), 0.5 mM of glutamine (GIBCO, Paisley, Scotland, UK), 50 U/mL of penicillin, and 50 µg/mL of streptomycin (GIBCO, Paisley, Scotland, UK). On the 4th day in vitro (DIV), the neuronal cultures were transduced with a mixture of rAAV viral particles carrying AAV-CAG-NES-GAF-CaMP3–sfGFP or AAV-CAG-NES-R-GECO1. Cells were imaged using an Andor XDi Technology Revolution multi-point confocal system on the 15th and 16th DIV.

Stimulation of the neuronal cultures was performed using a home-built electrical system as described earlier [13], except that the voltage pulses had an amplitude of ±70–120 V and the cultures were placed in a 24-well plate.

### 3.6. Statistics

To estimate the significance of the difference between two values, we used the Mann–Whitney rank sum test and provided *p*-values (throughout the text in parentheses) calculated for the two-tailed hypothesis. We considered differences as significant if the *p*-value was <0.05.

### 3.7. Ethical Approval and Animal Care

All methods for animal care and all experimental protocols were approved by the National Research Center “Kurchatov Institute” Committee on Animal Care (NG-1/109PR of 13 February 2020) and were in accordance with the Russian Federation Order Requirements N 267 МЗ and the National Institutes of Health Guide for the Care and Use of Laboratory Animals. Two C57BL/6 P0–1-old mice were used in this study. Mice were used without regard to gender.

## 4. Conclusions

In conclusion, we developed a version of the GAF-CaMP2–sfGFP NIR-GECI, called GAF-CaMP3–sfGFP, with a 1.9–2.9-fold enhanced dynamic range and compared its performance in cultured neuronal cells with the that of the red R-GECO1 indicator. GAF-CaMP3–sfGFP had a 1.88-fold lower molecular brightness as compared to the published data for NIR-GECO1 [3]. According to its excitation spectrum, only a 0.6 part of NIR-GECO1 is excited at 640 nm. Hence, the molecular brightness of GAF-CaMP3–sfGFP and NIR-GECO1 should be practically the same under excitation with lasers at 640 nm, which are commonly available in regular microscopes.

We modified the procedure for the screening of GAF-CaMP3 variants in bacterial systems and, accordingly, used its fusion with sfGFP to select mutants with better folding. However, this modification did not enhance the folding of the GAF-CaMP3 protein. Thus, we could not correct the main drawback of GAF-CaMP3 and it still demands sfGFP on its C-terminus for expression in mammalian cells.

In spite of the progress in the engineering of NIR-GECIs, calcium imaging using the GAF-CaMP3–sfGFP indicator in vivo in the mouse brain is challenging for researchers because of several obstacles. First, for one-photon imaging, we need to overcome technical problems concerning the modification of the optical path of commercially available NVista-like miniscopes. Second, the endogenous concentration of BV inside neuronal cells is not enough to obtain a strong signal (tacking into account the 6% molecular brightness of GAF-CaMP3–sfGFP vs. EGFP) and the development of the method for the delivery of external BV into the brain is necessary; co-expression of heme oxygenase cannot provide a significant increase in the brightness of NIR proteins in cells [16]. Alternatively, the development of a version of NIR-GECIs with an enhanced affinity to endogenous cellular BV might solve the latter problem.

## Figures and Tables

**Figure 1 ijms-21-06883-f001:**
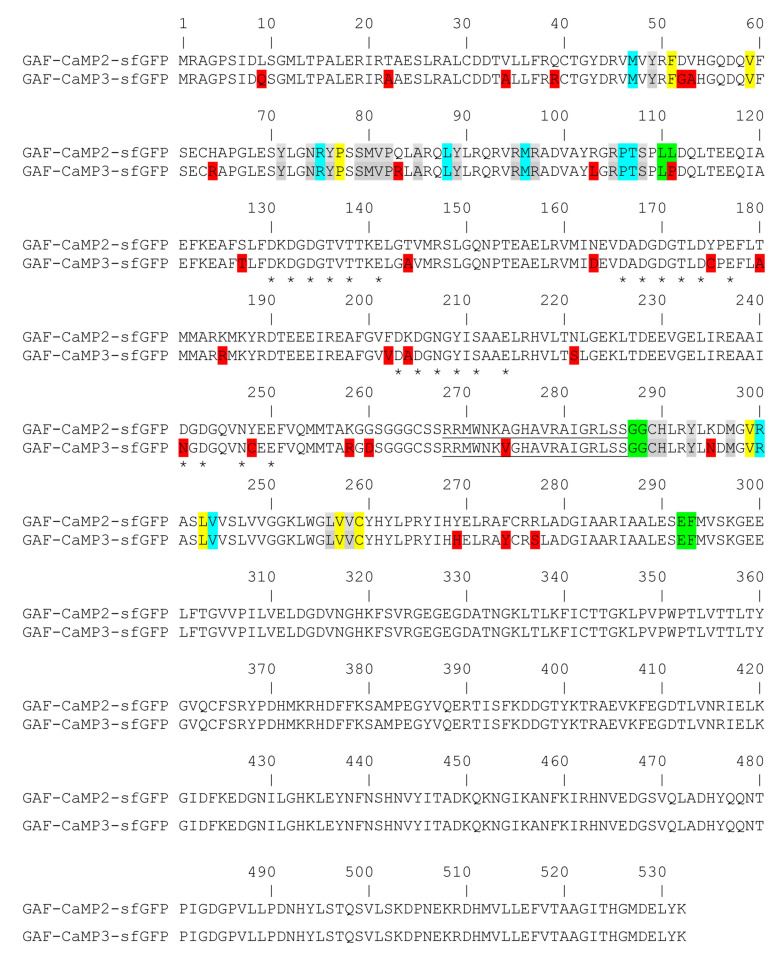
Alignment of the amino acid sequences for the GAF-CaMP3-sfGFP calcium indicator and its GAF-CaMP2–sfGFP progenitor. Alignment numbering follows that for the original GAF-CaMP2–sfGFP protein. Mutations in GAF-CaMP3–sfGFP related to the GAF-CaMP2–sfGFP progenitor, as well as the linkers between the fluorescent and calcium-binding domains, are highlighted in red and green colors, respectively. The residues within 4.5, 4.5–5.5, and 5.5–6.5 Å surrounding the biliverdin (BV) chromophore, as suggested according to the X-ray structure of PaBphP (PDB 3C2W), are highlighted in grey, cyan, and yellow colors, respectively. The residues in the CaM-part that assumed to bind Ca^2+^ ions are indicated with stars (*). The M13 peptide is underlined.

**Figure 2 ijms-21-06883-f002:**
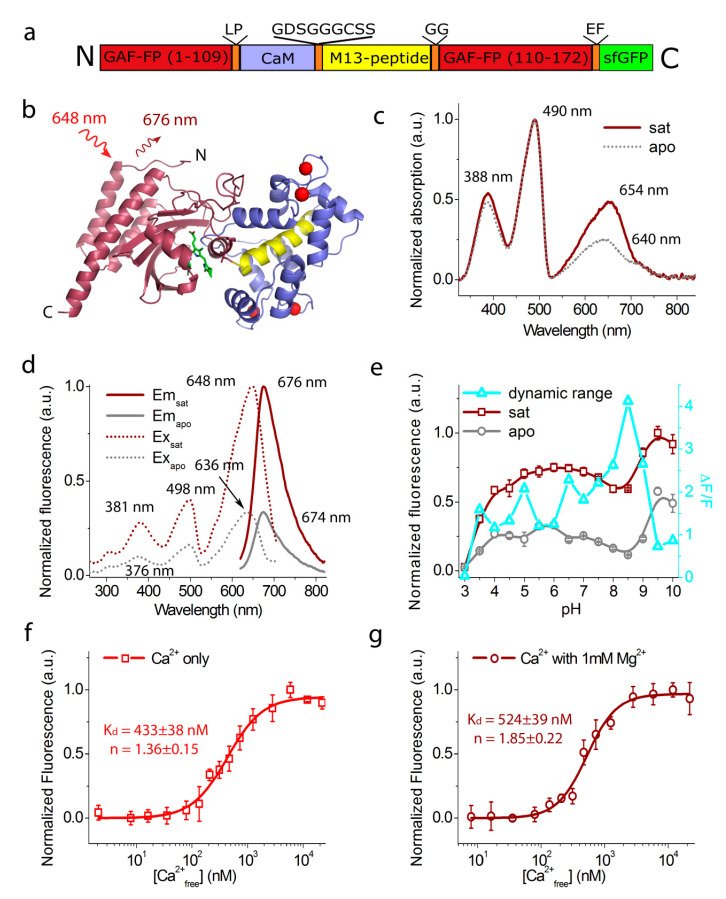
In vitro properties of the purified GAF-CaMP3–sfGFP indicator. (**a**) A schematic representation of the GAF-CaMP3–sfGFP indicator. (**b**) Suggested structural organization of the GAF-CaMP3 indicator based on the crystal structures of the GAF domain (PDB 3C2W) and CaM–M13 peptide (PDB 6XW2). The GAF domain, BV chromophore, CaM, M13 peptide, and calcium ions are highlighted in wine, green, cyan, yellow, and red, respectively. Excitation light and emission with maxima are shown by red and wine arrows, respectively. (**c**) Absorption spectra for GAF-CaMP3–sfGFP in Ca^2+^-bound and Ca^2+^-free states at pH 7.2. (**d**) Excitation and emission spectra for GAF-CaMP3–sfGFP in Ca^2+^-bound and Ca^2+^-free states at pH 7.2. (**e**) Fluorescence intensity for GAF-CaMP3–sfGFP in Ca^2+^-bound and Ca^2+^-free states and dynamic range as a function of pH. (**f**,**g**) Ca^2+^ titration curves for GAF-CaMP3–sfGFP in the absence (**f**) and in the presence (**g**) of 1 mM MgCl_2_ at pH 7.2. The experimental data were fitted by the Hill equation. (**e–g**) Three replicates were averaged for analysis. Error bars represent the standard deviation.

**Figure 3 ijms-21-06883-f003:**
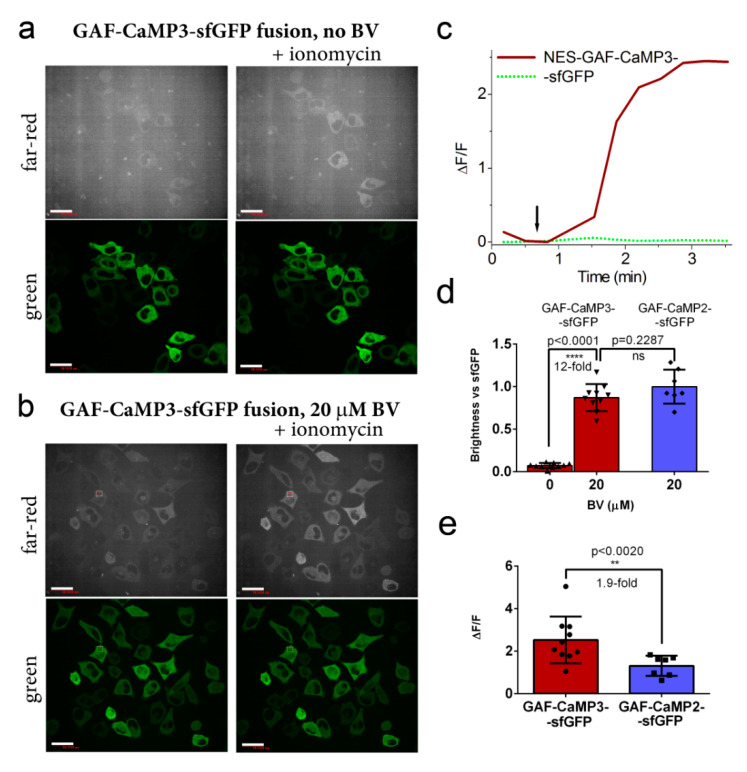
Optimization of the expression of the GAF-CaMP3–sfGFP indicator and its response to Ca^2+^ variations in HeLa cells. Confocal images of the HeLa cells expressing the nuclear export signal -(NES)-GAF-CaMP3–sfGFP fusion calcium indicator in the absence (**a**) and in the presence of 20 µM of external BV (**b**) before and after the addition of 2.5 μM of ionomycin. (**a**,**b**) Far-red and green fluorescence channels correspond to Ex 640 nm/Em 685/40 nm and Ex 488 nm/Em 525/50 nm, respectively. Scale bar, 50 µm. (**c**) The graph illustrates ΔF/F changes in green and far-red fluorescence of the GAF-CaMP3–sfGFP indicator in response to the addition of 2.5 μM of ionomycin. The changes in the graph correspond to the area indicated panel (**b**). (**d**) Comparison of averaged brightnesses for the GAF-CaMP3–sfGFP and GAF-CaMP2–sfGFP indicators in the HeLa cells upon the addition of 2.5 μM of ionomycin in the absence and in the presence of 20 µM of BV. (**e**) Comparison of the averaged ΔF/F responses for the GAF-CaMP3–sfGFP and GAF-CaMP2–sfGFP indicators in the HeLa cells upon the addition of 2.5 μM of ionomycin in the presence of 20 µM of BV. (**d**,**e**) Error bars are the standard deviations across 7–10 cells. The *p*-values show statistical difference between the respective values. *p* values ≤ 0.0001 and ≤ 0.01 are given four (****) and two (**) asterisks, respectively. Ns, not significant. BV (20 µM) was added 24 h before imaging.

**Figure 4 ijms-21-06883-f004:**
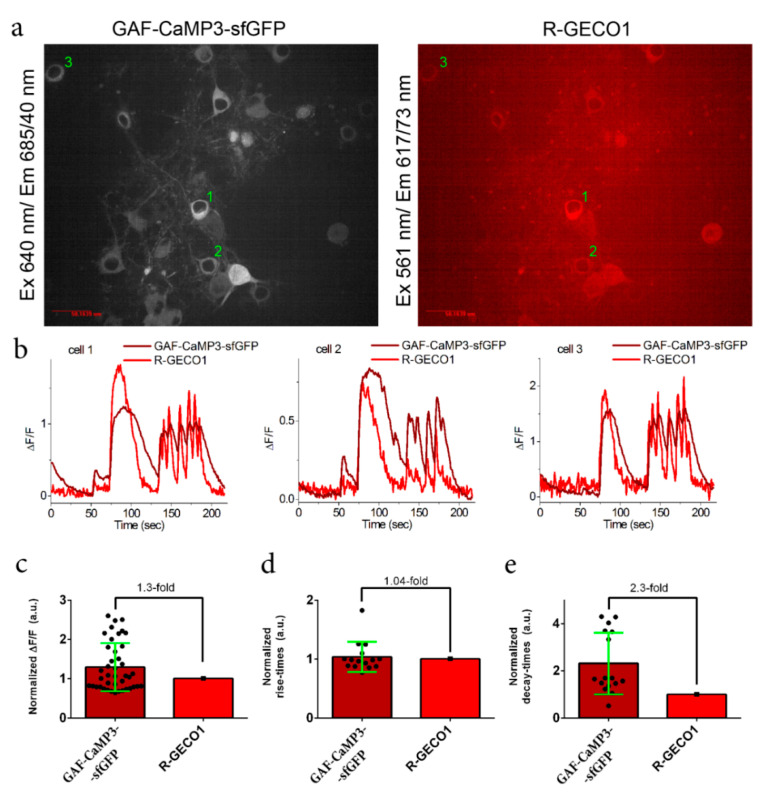
Two-color calcium imaging of the non-specific (spontaneous) activity of neuronal cultures co-expressing the near-infrared GAF-CaMP3–sfGFP and red R-GECO1 calcium indicators. (**a**) Confocal images of the neuronal cultures co-expressing the NES-GAF-CaMP3–sfGFP and NES-R-GECO1 indicators in the presence of 10 µM of external BV. Scale bar, 50 µm. BV (10 µM) was added 5–24 h before imaging. The neuronal cultures co-expressing the NES-GAF-CaMP3–sfGFP and NES-R-GECO1 indicators were imaged on the 15th day in vitro (DIV). The neuronal cultures were transduced on the 4th DIV with the mixture of recombinant adeno-associated viruses (rAAVs) carrying NES-GAF-CaMP3–sfGFP and NES-R-GECO1. (**b**) Examples of ΔF/F traces for the three cells marked on panel (**a**) as 1, 2, and 3 (see Appendix A for more traces). (**c**) ΔF/F responses of the GAF-CaMP3–sfGFP indicator normalized to the ΔF/F response of R-GECO1 in the same cell. The rise (**d**) and decay (**e**) times for GAF-CaMP3–sfGFP normalized to the respective times for R-GECO1 in the same cell. (**c–e**) Error bars are standard deviations across 5–8 cells and 15–42 spikes.

**Figure 5 ijms-21-06883-f005:**
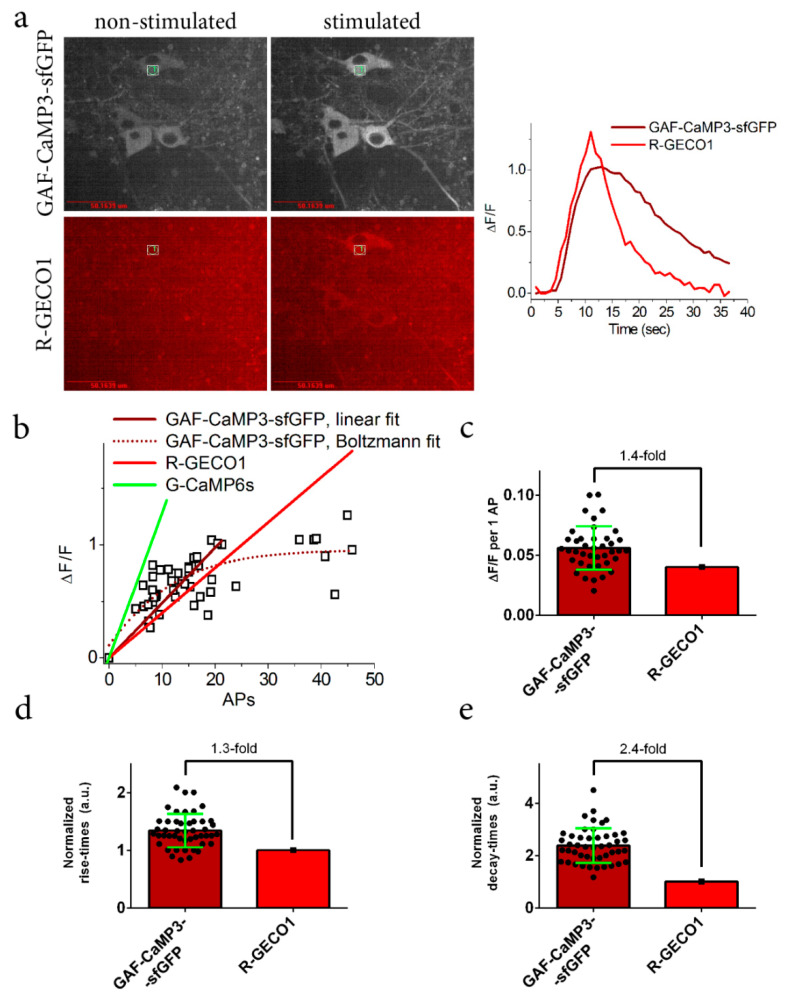
Comparison of the responses of the near-infrared GAF-CaMP3–sfGFP and the red R-GECO1 indicators to the external field stimulation of neurons co-expressing the genetically encoded calcium indicators (GECIs) in dissociated neuronal cultures. Neuronal cultures co-expressing the NES-GAF-CaMP3–sfGFP and NES-R-GECO1 indicators were imaged and stimulated on the 16th DIV. Neuronal cultures were transduced on the 4th DIV with a mixture of rAAVs carrying NES-GAF-CaMP3–sfGFP and NES-R-GECO1. (**a**) Confocal images of the neuronal cultures co-expressing the GAF-CaMP3–sfGFP and R-GECO1 indicators in the presence of 10 µM of external BV before and upon external electrical stimulation. The far-red (GAF-CaMP3–sfGFP) and red (R-GECO1) fluorescence channels correspond to Ex 640 nm/Em 685/40 nm and Ex 561 nm/Em 617/73 nm, respectively. Scale bar, 50 µm. BV (10 µM) was added 5–24 h before imaging. The graph illustrates the ΔF/F changes in far-red (GAF-CaMP3–sfGFP) and red (R-GECO1) fluorescence of the GAF-CaMP3–sfGFP and R-GECO1 indicators in response to the electrical stimulation. The changes on the graph correspond to the area indicated on the panel (**a**) as a circle labeled number 1. (**b**) Dependence of the ΔF/F responses of the GAF-CaMP3–sfGFP, R-GECO1 (linear fit) and GCaMP6s (linear fit, from [14]) indicators on a number of action potentials (APs). In case of GAF-CaMP3-sfGFP, the dependence was fitted by linear or Boltzmann equations in the range of 0–21 or 0–46 APs, respectively. A number of APs were calculated according to the ΔF/F response of the red R-GECO1 indicator co-expressing in the same cell, assuming that its ΔF/F value of 4 ± 1% corresponded to 1AP and had linear dependence from a number of APs [15]. (**c**) The ΔF/F response of the GAF-CaMP3–sfGFP indicator per AP was calculated in the range of 0–21 APs. The rise (**d**) and decay times (**e**) for GAF-CaMP3–sfGFP normalized to the respective times for R-GECO1 in the same cell. (**c**–**e**) Data were averaged across 40–47 cells, and the standard deviation (SD) is shown.

**Table 1 ijms-21-06883-t001:** In vitro properties of the GAF-CaMP3–sfGFP calcium indicator compared to the parental GAF-CaMP2–sfGFP indicator and smURFP and GAF-florescent protein (FP) permanently fluorescent proteins.

Properties			Proteins					
			GAF-CaMP3–sfGFP		GAF-CaMP2–sfGFP ^a^		smURFP ^a^	GAF-FP ^a^
			apo	sat	apo	sat		
Absorption maximum (nm)			640 (388, 490)	654 (388, 490)	630 (383, 490)	649 (383, 490)	642	637 (379)
Excitation maximum (nm)			636 (376, 498)	648 (381, 498)	630 (372, 493)	642 (374, 493)	642	635
Emission maximum (nm)			674 (514)	676 (514)	676 (514)	674 (514)	670	670
Quantum yield (%) ^b^			4.99 ± 0.06	7.69 ± 0.09	3.3 ± 0.1	6.9 ± 0.5	17.9 ± 0.2	7.3
ε (mM^−1^·cm^−1^) ^c^			14.4 ± 0.4	27.0 ± 0.7	15.7 ± 0.4	27.5 ± 0.9	180	49.8
Brightness vs. EGFP (%) ^d^			2.1	6.2	1.5	5.6	96	11
p*K*_a_			3.53 ± 0.05 9.18 ± 0.01	3.50 ± 0.05 8.9 ± 0.1	5.30 ± 0.02; 7.28 ± 0.02; ≥9.28 ± 0.06	4.89 ± 0.01; 7.60 ± 0.07; ≥9.31 ± 0.09	3.3	4.0; 7.8
ΔF/F (%)	Purified protein	0 mM Mg	197 ± 5		93 ± 11		NA	NA
		1 mM Mg	227 ± 11		78 ± 7			
	HeLa cells		252 ± 110		131 ± 47 *			
K_d_ (nM) ^e^	0 mM Mg		433 ± 38 (1.36 ± 0.15)		289 ± 29 (1.54 ± 0.22)			
	1 mM Mg		524 ± 39 (1.85 ± 0.22)		466 ± 50 (1.54 ± 0.23)			

^a^ Data from [4]. Data marked with an asterisk (*) were determined in this paper. NA, not applicable. ND, not determined. ^b^ Quantum yield (QY) for forms with an excitation maximum at 636−648 nm was determined at pH 7.20. smURFP was used as the reference standard. ^c^ Extinction coefficient for forms with an absorption maximum at 640−654 nm was determined relative to the sfGFP absorption at 490 nm [10]. ^d^ Brightness was calculated as a product of the quantum yield and extinction coefficient and normalized to the brightness of the enhanced green fluorescent protein (EGFP) that had an extinction coefficient of 56,000 M^−1^·cm^−1^ and a quantum yield of 0.6 [11]. ^e^ Hill coefficient is shown in parentheses.

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
