# Peer review of "GAF-CaMP3–sfGFP, An Enhanced Version of the Near-Infrared Genetically Encoded Positive Phytochrome-Based Calcium Indicator for the Visualization of Neuronal Activity"

_ijms, 2020, doi:10.3390/ijms21186883_

Round 1

Reviewer 1 Report

Subach and Subach generated a new form of near far red GECI that has similar brightness to it's progenitor GAF-GCaMP2-sfGFP but has 2.9 times larger dF/F0. This indicator was tested in HeLa cells and cultured neurons. This is a technically solid methodological paper with clear and honest conclusions and results. I have only minor suggestions:

  1. Abstract should contain a sentence or two regarding the comparison of GAF-GCaMP3-sfGFP to R-GECO in cultured neurons. This is, after all,  the most important result.
  2. Lines 101-117 - is too dense and difficult to follow for a non-expert in the area. Expand this part.
  3. Fig.3 a,b - scale bars missing
  4. Explain how you counted the action potentials or assumed the number of action potentials per stimulus. I appreciate this has been done many times and described in many publications but in my view the paper will benefit from much clearer explanation of this issue in the methods section. 
  5. Fig 5b will benefit from comparison with R-Geco and, perhaps, some forms of GCaMPs. I guess this data can be taken from previous papers, fits generated and added to the plot.
  6. Lines 317-326. The explanation why this construct cannot be used in2-p imaging is unclear. Elaborate and back with some references.  

Author Response

Response to Reviewer 1 Comments

We thank reviewer 1 for his review, valuable comments and useful suggestions, which we have addressed entirely in the revised manuscript.

Reviewer #1:

Subach and Subach generated a new form of near far red GECI that has similar brightness to it's progenitor GAF-GCaMP2-sfGFP but has 2.9 times larger dF/F0. This indicator was tested in HeLa cells and cultured neurons. This is a technically solid methodological paper with clear and honest conclusions and results. I have only minor suggestions:

Point 1:  Abstract should contain a sentence or two regarding the comparison of GAF-GCaMP3-sfGFP to R-GECO in cultured neurons. This is, after all,  the most important result.

Response 1: In the revised manuscript, Abstract, we added “In cultured neurons the GAF-CaMP3-sfGFP indicator showed linear DF/F response in the range of 0-20 APs and in this range demonstrated 1.4-fold larger DF/F response but 1.3- and 2.4-fold slower rise and decay kinetics, respectively, as compared to the same parameters for the R-GECO1 indicator.”.

Point 2:  Lines 101-117 - is too dense and difficult to follow for a non-expert in the area. Expand this part.

Response 2: To expand this part in the revised manuscript, lines 105-122, we added “GAF-CaMP3-sfGFP in apo- and sat-states has absorption maxima characteristic for Q-band of BV chromophore in bacterial phytochromes at 640 and 654 nm, respectively (Figure 2c). In addition to NIR absorbance at the major band (called the Q-band) phytochromes also absorb at the region about 400 nm (called the Soret band) which is a characteristic band for tetrapyrroles [9].”; “…molecular brightness (as a product of quantum yield and extinction coefficient)…”; “…in the presence of 1 mM Mg2+ (the condition, mimicking cytosolic cellular Mg2+ concentration),…”

Point 3:  Fig.3 a,b - scale bars missing

Response 3: In the revised manuscript, Figure 3a,b we added scale bars as white bands.

  • Point 4:  Explain how you counted the action potentials or assumed the number of action potentials per stimulus. I appreciate this has been done many times and described in many publications but in my view the paper will benefit from much clearer explanation of this issue in the methods section. 

Response 4: In the revised manuscript, in the legend to the Figure 5b, we mentioned that “A number of APs were calculated according to the DF/F response of red R-GECO1 indicator co‐expressing in the same cell, assuming that its DF/F value of 4±1% corresponds to 1AP and has linear dependence from a number of APs [14].”.

Point 5:  Fig 5b will benefit from comparison with R-Geco and, perhaps, some forms of GCaMPs. I guess this data can be taken from previous papers, fits generated and added to the plot.

Response 5: In the revised manuscript, Figure 5b, we added dependences of DF/F values versus APs for R-GECO and G-CaMP6s indicators.

In the main text, lines 214-215 we added “…but 2.7-fold lower vs GCaMP6s…”

Point 6:  Lines 317-326. The explanation why this construct cannot be used in2-p imaging is unclear. Elaborate and back with some references.  

Response 6: In the revised manuscript, “Conclusions” section, we deleted the sentence concerning possible problems with 2-p imaging of GAF-CaMP3-sfGFP, since we do not have data about GAF-CaMP3-sfGFP’s 2P-cross-section. Indeed, one-photon and two-photon spectra for GAF-CaMP3-sfGFP may be different according to the huge difference in 1P absorption and 2P cross-section for NIR-GECO1 (Figure 3a, Qian, Y. et al. Nature Methods, 2019).

Reviewer 2 Report

The topic undertaken by Authors is not new, however their work include some improvement to the gentically encoded near-inrared fluorescent calcium ion indicators. The manuscript is well organized and illustrated, thus I recommend it to be published.

Author Response

Response to Reviewer 2 Comments

We thank reviewer 2 for his review and recommendation for publication.

Reviewer #2:

The topic undertaken by Authors is not new, however their work include some improvement to the gentically encoded near-inrared fluorescent calcium ion indicators. The manuscript is well organized and illustrated, thus I recommend it to be published.
